# A Multi-Approach for In Silico Detection of Chromosome Inversions in Mosquito Vectors

**DOI:** 10.3390/microorganisms13102231

**Published:** 2025-09-24

**Authors:** Marcus Vinicius Niz Alvarez, Filipe Trindade Bozoni, Diego Peres Alonso, Paulo Eduardo Martins Ribolla

**Affiliations:** 1Pangene Laboratory, Biotechnology Institute, São Paulo State University (UNESP), Botucatu 18618-689, Brazil; marcus.alvarez@unesp.br (M.V.N.A.); filipe.trindade@unesp.br (F.T.B.); diego.p.alonso@unesp.br (D.P.A.); 2Genetics, Microbiology and Immunology Department, Bioscience Institute, São Paulo State University (UNESP), Botucatu 18618-689, Brazil

**Keywords:** mosquito, lcWGS, comparative genomics, structural variants

## Abstract

In Brazil, *Nyssorhynchus darlingi* stands out as the primary malaria vector. Chromosome inversions have long been recognized as critical evolutionary mechanisms in diverse organisms. In this study, we used biallelic SNPs to show that it is possible to detect chromosome inversions reliably with low coverage sequence data. We estimated chromosome inversions in an Amazon Basin sample of *Ny. darlingi* and compared them with *Anopheles gambiae* and *Anopheles albimanus* genomes in synteny analysis. The *An. gambiae* dataset benchmarked the inversion detection pipeline with known inversions. Genotyping by sequencing was performed using the LCSeqTools workflow for the lcWGS dataset with an average sequencing depth of 2x. A synteny analysis was performed for *Ny. darlingi* inversions regions with *An. gambiae* and *An. albimanus* genomes. The sliding window analysis of PCA components revealed 10 high-confidence candidate regions for chromosome inversions in *Ny. darlingi* genome and two known inversions for *An. gambiae* with possible identification of breakpoints and adjacent regions at lower resolution. We demonstrate that lcWGS is a cost-effective and accurate method for detecting chromosome inversions. We reliably detected chromosome inversions in *Ny. darlingi* from the Brazilian Amazon that does not share similar inversion arrangements in *An. gambiae* or *An. albimanus* genomes.

## 1. Introduction

Malaria is a disease caused by *Plasmodium* parasites and transmitted through the bites of infected Anophelinae mosquitoes. Despite ongoing efforts, it remains a major global health challenge, particularly in tropical and subtropical regions [1]. Several factors contribute to malaria transmission, including socio-ecological factors and changes in the human host and vector behavior. In 2022, there were an estimated 249 million malaria cases globally, exceeding the pre-pandemic level of 233 million in 2019 by 16 million cases [2]. Increased human population and land use/land cover change (LULC) influence the biological community, including Anophelinae mosquitoes, particularly those with some degree of synanthropy and competence to transmit *Plasmodium* sp. that circulate in the Amazon region [3]. This vast region is responsible for 99.5% of human malaria in Brazil, mainly *Plasmodium vivax* (>90% in 2019) [4].

Among the diverse Anophelinae mosquitoes, *Nyssorhynchus darlingi* is the primary vector responsible for malaria transmission in the Neotropical region, including the Amazon basin [5]. This vector demonstrates high anthropophilic behavior in many areas, sometimes combined with opportunistic zoophilic (non-human) feeding [6]. *Ny. darlingi* presents a chromosomal composition of 2*n* = 6, with two pairs of autosomes: the largest (III) is submetacentric and the smallest (II) is metacentric. For sex chromosomes, X is acrocentric and Y is punctiform [7]. The distribution of the species reaches from Southern Mexico to Northern Argentina and from the East of the Andean Mountains to the coast of the Atlantic Ocean. Current strategies of integrated vector management, including the widespread use of insecticide-treated bed nets and the regular application of indoor residual spraying, are important for malaria control; however, they may still be insufficient to completely eliminate transmission in all endemic regions, and therefore the development of new, complementary strategies remains important in this context. Understanding the genetic basis of vectorial capacity in *Ny. darlingi* is essential for developing effective control strategies to combat malaria transmission [8].

Chromosome inversions, structural rearrangements where a chromosome segment is reversed in orientation, have long been recognized as critical evolutionary mechanisms in diverse organisms [9]. In mosquitoes, chromosome inversions have been associated with ecological adaptation, population divergence and speciation. These inversions can suppress recombination, leading to the maintenance of coadapted gene complexes and potentially facilitating the rapid evolution of traits related to vectorial capacity, such as insecticide resistance and host preference [9]. In *Anopheles gambiae*, two inversions (2La and another in the 2R arm) have been associated with important physiological and transcriptional differences, depending on sex, climate and epistatic interactions, demonstrating the adaptive effect of these inversions [10]. The 2La inversion is directly linked to desiccation resistance, with strong implications for survival strategies in arid environments [11,12]. In *Anopheles funestus*, a study in Africa suggested that chromosomal inversions promote ecological differentiation even when neutral markers fail to detect significant population structure [13]. Previous studies have identified chromosomal inversions in *Ny. darlingi* populations, particularly in the Amazon region, lack a comprehensive understanding of their prevalence, distribution and functional significance [8].

The rapid evolution of technologies involved in whole-genome sequencing (WGS) has precipitated substantial reductions in the per-base sequencing cost. Nonetheless, large-scale projects that require the sequencing of large sample sizes remain financially burdensome, potentially presenting insurmountable obstacles within specific laboratory contexts. One economically viable strategy entails genotyping-by-sequencing for low-coverage WGS (lcWGS), concomitant with imputation procedures that furnish adequate genomic data for precise marker selection [14]. Although the fidelity of variant detection diminishes in genomes characterized by shallow coverage depth, thereby engendering a heightened false-positive rate, this limitation is ameliorated through amalgamating information across multiple samples, thus augmenting the discernment of common variants [14,15]. Notably, imputation-driven genotype inference has been empirically validated for both panel-based genotyping and sequencing genotyping modalities [16], facilitating the potential adoption of extremely low-coverage WGS (EXL-WGS) for identifying variants at a markedly diminished expenditure relative to conventional WGS methodologies [17,18]. Li and collaborators have demonstrated that detecting rare variants in LcWGS samples poses considerable challenges due to the intrinsic difficulty in distinguishing authentic rare alleles from sequencing artifacts [19]. Notably, the enumeration of variants positively correlates with the prevalence of polymorphisms among the sequenced individuals within the delineated population subset. Given the diverse methodological approaches available for EXL-WGS analysis, meticulous calibration of each method’s sensitivity is imperative, given that the attenuation in coverage inevitably amplifies the potential for erroneous identifications.

In the present study, we used lcWGS markers to investigate the population of *Ny. darlingi* collected in Mâncio Lima, Acre state, Brazil. We conducted a comprehensive genomic analysis of chromosome inversions in *Ny. darlingi* and performed a comparative genomics analysis with *Anopheles albimanus* and *Anopheles gambiae* genomes. This study examines closely related *Anopheles* species, providing insights into the evolutionary dynamics across different mosquito lineages. Our findings contribute to the broader understanding of mosquito genetics, evolution and vector biology, with implications for malaria control strategies and vector management programs.

## 2. Materials and Methods

### 2.1. Sequencing Data Acquisition

This study used publicly available sequencing data from the National Center for Biotechnology Information (NCBI). Sequencing data types were selected based on their relevance to ensure comprehensive analysis. Sequencing data from lcWGS data of 321 *Nyssorhynchus darlingi* larvae and adults were obtained from the Bioproject PRJNA683015. The samples used in this study were collected in Mâncio Lima, Acre, Brazil.

### 2.2. Genome Reference

*Ny. darlingi*, *An. gambiae* and *An. albimanus* reference genomes used are publicly available at the NCBI database with respective accession numbers in Table 1. All reference genome assemblies are at chromosome level resolution, providing contiguous sequences suitable for downstream genomic analysis. 

### 2.3. Genotyping by Sequencing and Variant Calling

The genotyping by sequencing pipeline was performed using LCSeqTools v0.1.0 [20]. Through the LCSeqTools, different steps are automatically applied optimized for lcWGS data, with customizable predefined parameters for filtering and quality control. Firstly, reads are trimmed using the Trimmomatic v0.39 software [21] step, with trimming parameters as follows: headcrop = 10, trailing = 20 and minlen = 100. Trimmed reads alignment was performed with the Burrows-Wheeler Aligner v0.7.17 software package [22], this step applies the *bwa mem* method with its default parameters for single-end mapping, followed by variant calling with the SamTools v1.10 software package [19] using the *bcftools call-m* method with its default calling penalties and weighting parameters. The LCSeqTools estimates missing data rates as the ratio of truly missing data, defined as missing genotype normalized probability (PL) fields due to zero depth, to non-missing data, defined as non-missing PL fields with at least one read depth. Given that, variant filtering parameters applied before genotype imputation were as follows: minor allele frequency (MAF) < 0.1, max missing data per sample and per variant < 0.5, genotype sequencing depth < 5 and genotype quality < 20. LCSeqTools uses depth threshold for genotype omission, so that GT fields are set as missing but PL are retained if non-missing. Using LCSeqTools, the last step applied genotype imputation with the BEAGLE v4.1 software package [23] using the PL method, with its imputation algorithm parameters as default and no reference panel. No post-imputation filtering through LCSeqTools was applied.

The genome coverage statistics was calculated using the bedtools v2.30.0 [24] software package with the *genomecov* method using the resulting BAM files from the read mapping step. A final filtering step was applied to the variant dataset with the SNP pruning algorithm from the PLINK v1.9 software package [25], using the *indep-pairwise* method, window size = 14 kbp, step size = 9 kbp and 0.05 as *r*^2^ threshold, as Alvarez and collaborators showed that the expected *r*^2^ value at a 12.57 kbp distance is approximately 0.1 in this *Ny. darlingi* population [20].

### 2.4. Data Processing and Statistical Analysis

All data manipulation, analysis and plotting were performed using RStudio Server (Ocean Storm version 2023.12.0) and R Language v4.2.1 [26], including packages provided by the tidyverse metapackage [27].

### 2.5. Chromosome Inversion Identification

Multiple approaches were combined to check the presence of chromosome inversion signals. Principal component analysis (PCA) for each chromosome was performed using PLINK, and the sliding window variance of SNP weight (eigenvector) estimates for each principal component were calculated within 10 kbp sliding windows with step size of 7.5 kbp, avoiding excessively low *r*^2^ SNPs in the window. Based on the absolute PCA SNP weights, the top 1% SNPs for each component were retained and the Identical by State (IBS) matrix was calculated using the respective SNP subset. The multidimensional scaling (MDS) technique was applied to the Euclidean IBS matrix with the cmdscale function from R base (k = 1 for genotype clustering and k = 2 for pairwise comparisons) and, subsequently, a clustering technique with the cmeans function from the e1071 package for R [21]. Only components clustered into three well-defined groups were retained, later identified as AA, AB and BB inversion genotypes using cmeans maximal membership probability.

A genome-wide association test was applied for each candidate chromosome inversion, using the linear model from Plink software, considering AA, AB and BB samples as quantitative 0, 1 and 2 phenotypes, respectively. A Bonferroni post hoc test was applied for multiple comparisons correction and SNPs were considered statistically significant when the adjusted *p*-value < 0.05.

A pairwise SNP linkage disequilibrium estimate analysis was performed for SNP pairs within ranges of 12 kbp of distance and the sliding window *r*^2^ median was calculated for each chromosome within sliding windows of 0.5% of the respective chromosome length. The Spearman correlation coefficient ρ was estimated for *r*^2^ and the mean SNP absolute weights using the cor.test function from R and was considered statistically significant when *p*-value < 0.05.

### 2.6. Chromosome Correlation Tests and Association Tests

All chromosome inversions were submitted to a pairwise Spearman correlation test using the estimated sample genotypes data and the *p*-values were adjusted for multiple comparisons using the Benjamini–Hochberg False Discovery Rate method (FDR). Correlations were considered statistically significant when FDR-adjusted *p* ≤ 0.05.

### 2.7. Pipeline Validation with Known Variants for Anopheles gambiae

High-coverage genome-wide variant data from the first phase of the “The *Anopheles gambiae* 1000 Genomes Consortium”, publicly available at EMBL-EBI, were used as reference (ERZ373588 with GCF_000005575.2 genome reference). An in silico lcWGS dataset was generated for chromosome identification pipeline validation, matching results with known *An. gambiae* structural variants (AgamP4 variant panel publicly available at Ensembl Metazoa). AgamP4 structural variants were localized in the GCF_000005575.2 genome reference using Minimap2 v2.25 [28]. A subset of 200 samples from the Republic of Cameroon group was selected from the full variant panel and lcWGS data were generated with the art_illumina program from the ART v2.5.8 software package [29] with approximately 2x sequencing depth per sample.

### 2.8. Comparative Genomics Analysis

An orthology inference was performed for *An. gambiae* and *An. albimanus* with *Ny. darlingi* as reference using BLASTp v2.12.0 [30] and the respective proteomes through the accession codes. The OrthoFinder v2.5.5 software [31] was also used as a second approach for ortholog inference and convergence check with BLASTp results. A genome synteny analysis was performed between the *An. gambiae*, *An. albimanus* and *Ny. darlingi* genomes using MCScanX v1.0.0 [32].

## 3. Results

### 3.1. Ny. Darlingi Chromosome Inversions Detection

The LCSeqTools pipeline generated a variant panel containing 4,241,254 SNPs for the *Ny. darlingi* sample. The sample-wise median proportion of genomic positions with zero coverage, representing missing data, was 36.89% (Q1 = 18.09%, Q3 = 57.94%, min = 2.73%, max = 76.04%). Also, at 5× depth (no GT omission threshold), the sample-wise observed median was 3.48% (Q1 = 1.11%, Q3 = 12.79%, min = 0.24%, max = 81.10%). The full sample-wise genome coverage statistics after properly read alignment are shown in Appendix A. After the SNP pruning, Plink retained 626,409 SNPs, around 3.4 per Kbp.

The PCA components analysis revealed continuous high-shifted regions from SNP weight sliding windows variance estimates, counting 10 candidate regions: 2, 4 and 4 shifted regions for chromosomes 2, 3 and X, respectively (Figure 1). Clustering analysis of each region revealed three well-defined genetic clusters with approximately the same 0.5 genetic distance between samples AA at the leftmost and AB in the middle and AB and BB at the rightmost (Figure 2). The chromosome inversion genotype membership probabilities of each sample are provided in the Appendix A. Genome-wide association test results for each remaining component showed statistically significant SNPs concentrated in converging chromosome regions of its respective PCA shifted variance windows (Figure 1). The chromosome inversion coordinate estimates are represented in Table 2.

The SNP pairwise linkage disequilibrium analysis showed that, despite the median *r*^2^ within a distance of 12 to 13 kbp being 0.08, 0.06 and 0.12 for chromosomes 2, 3 and X, respectively, spikes reaching almost double the median *r*^2^ were found in different regions along the chromosomes (Figure 1). Significant correlations were observed between *r*^2^ and mean SNP absolute weight from PCA components, as the ρ estimates were 0.62, 0.43 and 0.58 for chromosomes 2, 3 and X, respectively.

### 3.2. In Silico Validation of the Pipeline

The *An. gambiae* lcWGS simulated data resulted in a variant panel containing 3,825,409 SNPs. The simulated genotypes accuracy after genotyping by sequencing pipeline and imputation showed 84.52% and 92.02% for average genotype concordance and average allelic concordance, respectively (Table 3). Also, the total dosage R^2^ was 0.7465. Plink retained 658,413 SNPs after the pruning step, resulting in 2.36 SNPs per Kbp. Two meaningful components were detected with the sliding window PCA weight analysis approach and, subsequently, a genome-wide association test revealed statistically significant SNPs concentrated within two regions, as coordinate estimates were [20,466,374, 42,223,401] and [19,275,187, 26,282,153] for 2L and 2R (Figure 3). Both inversion estimates greatly coincided with the well-known *An. gambiae* inversions: 2La [20,524,057, 42,165,532] and 2Rb [19,023,924, 26,758,676]. The observed range offsets were −57,683 and 57,869 for the start and end coordinates of 2La, and 251,263 and −476,523 for 2Rb. Significant correlations were observed between *r*^2^ and mean SNP absolute weight from PCA components, as the ρ estimates were 0.88 and 0.49 for 2L and 2R, respectively.

### 3.3. Genome Synteny Analysis

Comparative genomics analysis revealed similar arrangements between *Ny. darlingi* and *An. albimanus* genomes and a significant rearrangement when comparing chromosomes 2 and 3 from *An. gambiae* and *Ny. darlingi* (Figure 4). Synteny analysis for each chromosome inversion region showed no similarity between *Ny. darlingi* detected inversions and *An. gambiae* known chromosome inversions (Figure 5). No data was available comparing *An. albimanus* inversions.

## 4. Discussion

This study shows that lcWGS provides a powerful and cost-effective approach for detecting chromosome inversions in large sample sizes. The method offers a practical alternative to high-coverage sequencing, substantially reducing the overall study budget while resulting in a lower resolution but reliable process of inversion presence detection in the population. However, it is important to note that the sensitivity of this approach was not estimated because no chromosome inversion genotyping metadata is available for both datasets samples to be used as reference, so further tests should be designed to estimate the performance on lcWGS data. The in silico experiment reinforces that, besides the higher genotyping error probability of lcWGS, high-confidence inversion signal detection is possible because multiple independent SNPs are reduced into a single genotype data with the PCA technique, reducing any lcWGS genotyping error and false-positive SNP bias. The genotype association analysis could extract an approximation of the chromosome inversion coordinates using the *p*-value threshold, but with unpredictable offsets from the true chromosome inversion ranges.

Nowling and collaborators used a similar approach to traditional WGS data for *An. gambiae* dataset and found a comparable performance for PCA-based analysis on chromosome inversion detection [33]. Although there are various pipelines for detecting chromosome inversion using SNPs or different genetic parameters as they summarized, a simple sliding window analysis is enough to detect inversion footprints if an in-depth eigenvectors analysis is performed. The *An. gambiae* inversion range estimates from lcWGS data in this study were consistent and close to the known ranges [34]. Even on genome drafts in which chromosome data is not well assembled, it is possible to use PCA clustering to detect inversion presence because eigenvectors are estimated essentially with a set of independent SNPs, so contiguity is not needed for detection. This is an essential feature for non-model organism studies, but the more shattered the assembly is, the more difficult it will be to estimate the inversion coordinate range.

A significant correlation is observed between linkage disequilibrium and the absolute values of PCA eigenvectors, although PCA-based methods provide a more precise approach for detecting structural variants [35]. The observed linkage disequilibrium oscillation towards the inversion breakpoints is a known effect caused by the change in the recombination rate [36]. Therefore, our main goal with the proposed approach is to identify the most informative SNPs for chromosome inversions, which are mainly composed of non-recombinant sites and almost-fixed alleles. The methodology is capable of detecting those fixed SNPs associated with inversions. The exact location of breakpoints is not relevant. By focusing on the most informative, high-confidence SNPs, this approach enables robust and clear interpretability of chromosome inversion presence, providing a practical tool for studies of genomic structural variation and population genomics.

The genome synteny analysis revealed an interesting rearrangement between the arms of chromosomes 2 and 3 for *Ny. darlingi* and *An. albimanus* when compared to *An. gambiae* genome. This phenomenon is evidenced by other comparative genomics studies focusing on Anophelinae, which have consistently reported analogous chromosomal rearrangements [37,38]. The ten chromosome inversions detected in *Ny. darlingi* genome did not show similar ranges or arrangements for known *An. gambiae* inversions in Figure 5. However, *An. albimanus* revealed some interesting equivalent syntenic regions for the chromosome inversions. As far as we know, there is no data available on *An. albimanus* chromosome inversions. Unfortunately, during this study, there was no publicly accessible sequencing data for *An. albimanus* that would facilitate a comprehensive analysis of inversion footprints using the methodology employed in this study.

Soboleva and collaborators also identified two large nested X chromosome inversions using physical genome mapping approach with fluorescence in situ hybridization of gene markers in polytene chromosome to locate the breakpoints for *Anopheles atroparvus* and *Anopheles messeae*. Thus, our approach is not directly comparable to their method, both in data type or resolution, the reported ranges seem slightly different from the observed in this study [39]. Their findings provide complementary evidence on X chromosome inversions for Anophelines.

## 5. Conclusions

Chromosomal inversions are important biological markers for genome evolution, environmental adaptation and speciation. Our study showed that lcWGS is particularly promising for large-scale genomic studies, offering a cost-effective alternative strategy to detect chromosome inversions. On the other hand, a 2× sequencing depth will increase genotyping errors, resulting in a limited precision for the exact location of the breakpoints.

The sliding window analysis of PCA eigenvectors to detect inversions ensures accurate inversion detection and range estimation with a simple, standalone approach. Ten chromosome inversions were found using lcWGS data from *Ny. darlingi* samples collected in one municipality located in the Brazilian Amazon basin. The synteny analysis revealed that *Ny. darlingi* do not share similar chromosomal inversion arrangements with *An. gambiae*. This finding emphasizes the 100 MYA separation of *Ny. darlingi* and *An. gambiae* represented by the Pangea breakup. The lack of data on *An. albimanus* chromosome inversions impairs any comparisons with *Ny. darlingi.*

The methodology presented in this paper facilitates the detection of chromosomal inversions based on a low coverage data and enables population genetics studies to be performed, mitigating the influence of chromosomal inversions.

## Figures and Tables

**Figure 1 microorganisms-13-02231-f001:**
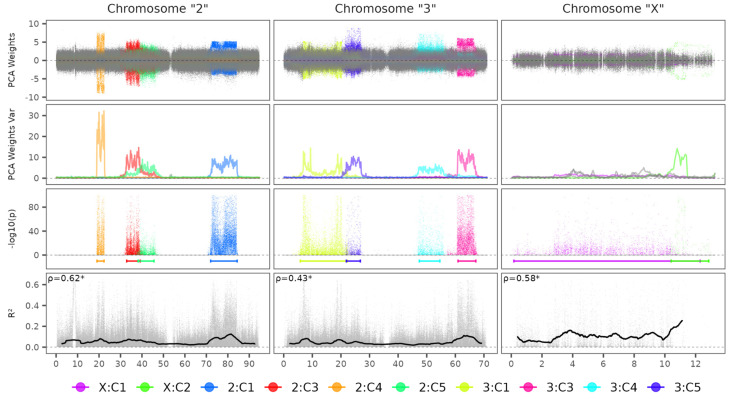
Chromosome inversion detection analysis for *Ny. darlingi* dataset. PCA Weights Var: Sliding window of SNPs weights variance estimate. ρ: Spearman correlation coefficient for *r*^2^ and mean absolute SNP weights. *: Correlation test *p* value < 0.05. Legend stands for Chromosome: Principal Component. Horizontal colored segments represent the inversion coordinate estimates described in Table 2. Horizontal scale in Mbp.

**Figure 2 microorganisms-13-02231-f002:**
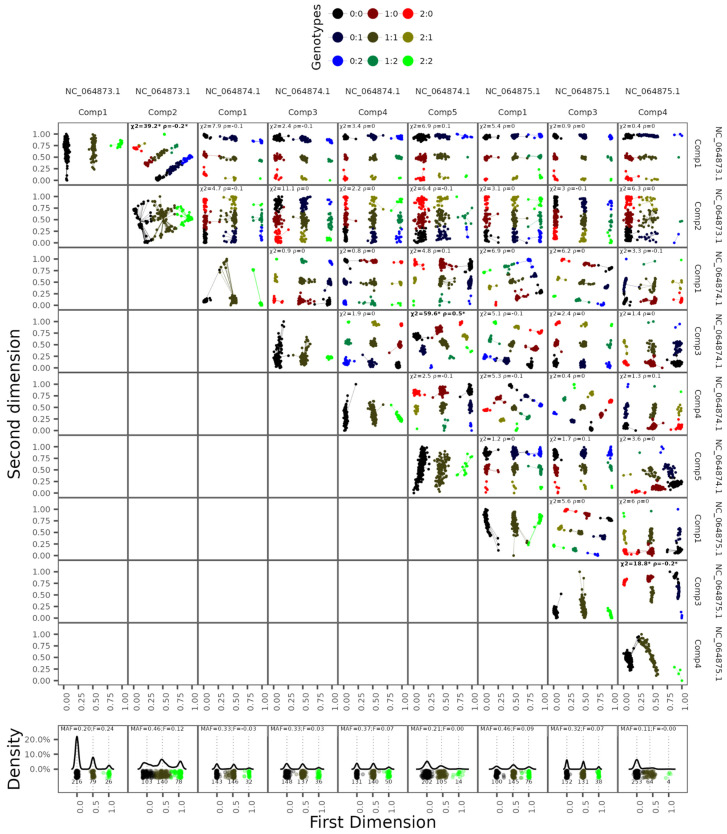
Inversion genotyping analysis. Top biplots are displayed in two dimensions (MDS with k = 2), showing the relative distances between samples based on genetic similarity from the top 1% most representative SNPs for each respective inversion. *χ*^2^: Chi-square test between genotypes from two respective inversions. ρ: Spearman correlation coefficient for genotypes states of both inversions. Bottom line plots represent in one dimension the density of samples grouping for each genetic cluster for each inversion. Numbers under cluster points represent cluster counts. MAF: Minor inversion frequency. F: Inversion fixation index. *: FDR adjusted *p* values < 0.05.

**Figure 3 microorganisms-13-02231-f003:**
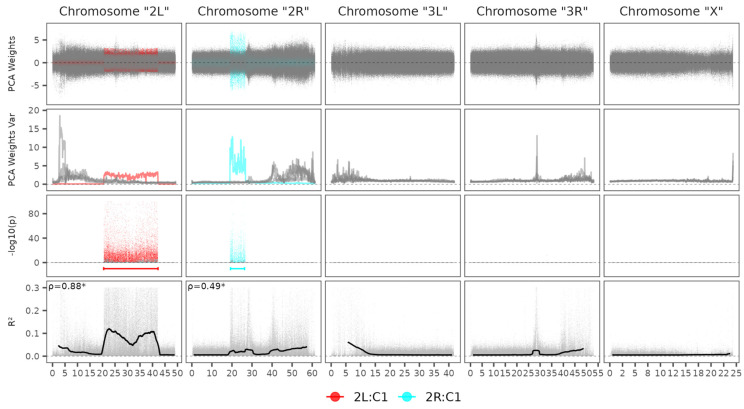
Chromosome inversion detection analysis for *An. gambiae* dataset. PCA Weights Var: Sliding window of SNPs weights variance estimate. ρ: Spearman correlation coefficient for *r*^2^ and mean absolute SNP weights. *: Correlation test *p* value < 0.05. Legend stands for Chromosome: Principal Component. Horizontal scale in Mbp.

**Figure 4 microorganisms-13-02231-f004:**
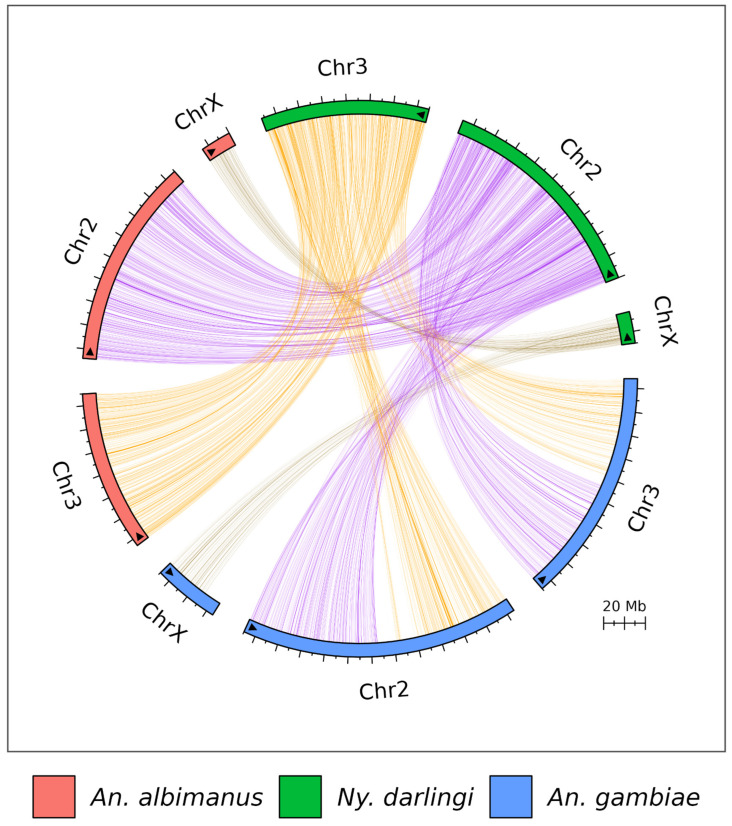
Genome synteny analysis for *Ny. darlingi* with *An. gambiae* and *An. albimanus*. Black arrow inside each segment represents the original reference orientation.

**Figure 5 microorganisms-13-02231-f005:**
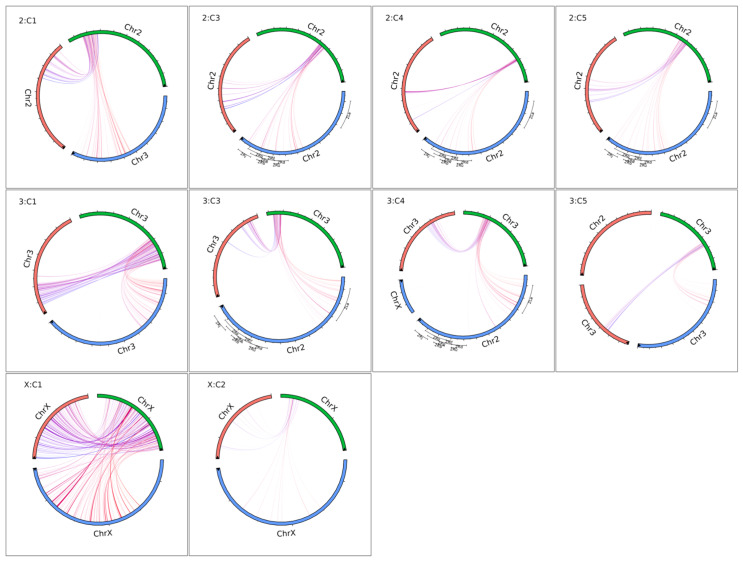
Chromosome inversion synteny analysis. Green color: *Ny. darlingi*. Blue color: *An. gambiae*. Red color: *An. albimanus*. Upper left label indicates the chromosome and the i-th principal component in which chromosome inversion was detected for *Ny. darlingi*. Known *An. gambiae* inversions for chromosome 2 are represented with the black line segments outside the circles. Black arrow inside each segment represents the original reference orientation. Only chromosomes with at least one link were retained in each plot.

**Table 1 microorganisms-13-02231-t001:** Summary of reference genomes used in the study.

Specie	Reference Accession	Number of Chromosomes	Genome Size	Scaffold N50 (L50)
*Nyssorhynchus darlingi*	GCF_943734745.1	3	181.6 Mb	95 Mb (1)
*Anopheles gambiae*	GCF_943734735.2	3	264.5 Mb	99 Mb (2)
*Anopheles albimanus*	GCF_013758885.1	3	172.6 Mb	89 Mb (1)

Accession codes can be used to retrieve the corresponding records through the NCBI.

**Table 2 microorganisms-13-02231-t002:** *Ny. darlingi* chromosome inversion coordinate estimates based on genome-wide association test.

Chromosome	Accession Code	Component	Start	End	Length
2	NC_064874.1	C1	71,998,919	84,371,386	12,372,467
2	NC_064874.1	C3	32,875,093	39,167,094	6,292,001
2	NC_064874.1	C4	19,048,046	22,354,438	3,306,392
2	NC_064874.1	C5	38,207,551	45,614,642	7,407,091
3	NC_064875.1	C1	5,786,821	22,248,013	16,461,192
3	NC_064875.1	C3	60,944,934	67,169,663	6,224,729
3	NC_064875.1	C4	47,375,561	54,594,956	7,219,395
3	NC_064875.1	C5	21,879,305	26,791,351	4,912,046
X	NC_064873.1	C1	169,219	12,292,796	12,123,577
X	NC_064873.1	C2	10,405,329	12,856,594	2,451,265

Component: Principal component from PCA.

**Table 3 microorganisms-13-02231-t003:** Post LCSeqTools imputation performance for Simulated lcWGS data.

MAF	Genotype Concordance	Allelic Concordance	Dosage R^2^	*N*	N FP	FP Rate
(0, 0.1]	92.48%	96.19%	0.8482	14,082	-	-
(0.1, 0.2]	90.80%	95.28%	0.8227	1,097,632	8557	0.7736%
(0.2, 0.3]	85.27%	92.40%	0.7537	1,161,704	141	0.0121%
(0.3, 0.4]	80.67%	90.02%	0.6898	830,122	158	0.0190%
(0.4, 0.5]	77.89%	88.64%	0.6379	712,671	342	0.0480%

MAF: Minor allele frequency range based on reference data. FP: False-positive SNPs; Genotype Concordance: Average proportion of genotypes that are exactly identical between imputed low-coverage simulated data and reference data; Allelic Concordance: Average proportion of shared alleles between genotypes from imputed low-coverage simulated data and reference data. Dosage R^2^: Squared Pearson correlation of alleles dosages. No false positives were identified with MAF (0, 0.1] because simulated data is filtered for MAF < 0.1 and true positives in that range stand for inflated frequencies (simulated MAF > 0.1) compared to the reference frequencies.

## Data Availability

The original data presented in the study are openly available in Zenodo, record 13755768 (https://doi.org/10.5281/ZENODO.13755768) and Zenodo, record 17064546 (https://doi.org/10.5281/zenodo.17064546).

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
