# Peer review of "A Multi-Approach for In Silico Detection of Chromosome Inversions in Mosquito Vectors"

_microorganisms, 2025, doi:10.3390/microorganisms13102231_

Round 1

Reviewer 1 Report

Comments and Suggestions for Authors

The authors present a multi-method study with low-coverage whole-genome sequencing (lcWGS; ~2×) to detect chromosome inversions in a large sample of Nyssorhynchus darlingi (n = 321), test their pipeline with an in silico simulation of Anopheles gambiae lcWGS, and align inversion regions by synteny with An. gambiae and An. albimanus. The approach strengths include having a large sample size for lcWGS, conducting a simulation/benchmark phase on a species known to harbor inversions, incorporating complementary analyses (PCA-weight sliding windows, IBS/MDS clustering, LD analyses, GWAS of inferred genotypes), and providing analysis scripts through Zenodo. Weaknesses are the lack of methodological critical detail and quantitative validation criteria (imputation accuracy, genotype concordance, breakpoint confidence intervals), too high and unjustified filtering thresholds that conflict with stated sequencing depth, and suboptimal reporting of cohort/sample QC and potential confounders in association analysis. Overall quality: 80/100. Quality of language: 7/10 — mostly comprehensible with some grammatical mistakes, some misphrasing, and places where methodological explanation is lost through concise phrasing.

The Methods outline the core software packages employed (LCSeqTools, BWA, samtools, BEAGLE 4.1, PLINK v1.9, OrthoFinder, MCScanX) and the most important algorithmic steps (read alignment, variant calling, hard filters, imputation, PCA variant-weight sliding window, IBS→MDS→cmeans clustering, GWAS under 0/1/2 genotype treatment, LD r² sliding windows, synteny/orthology using BLASTp + OrthoFinder). But the section omits numerous significant parameters and validation processes needed to test for reproducibility as well as the validity of the conclusions:

Sample- and site-level QC information are missing. The author does mention variant filters (MAF < 0.1, max missing < 0.5, genotype depth < 5, GQ < 20) but does not report per-sample mean depth, per-site missingness distribution, number of samples removed due to low depth or too much missingness, or whether PCR duplicates or contamination were assessed. With lcWGS (~2×) a genotype sequencing-depth filter of <5 is semantically inconsistent: it implies most genotypes are filtered except for imputation—this must be explained (e.g., were raw genotype calls retained only for imputation input and then masked?). Provide per-sample depth histogram, per-site missingness, and number of sites/samples filtered by step. Variant-calling pipeline reproducibility and versions: "samtools" and "BWA" are mentioned but specific versions, command-line flags, mapping quality cut-offs, and realignment/indel calling are not specified. LCSeqTools v0.1.0 is mentioned but not its default parameters. These all have an important effect on low-coverage call sets and need to be provided (commands or a workflow file).

Imputation details are not fully provided. BEAGLE 4.1 was run with PL field, but there is no reporting of the reference panel (if any) used, whether phasing was performed prior to imputation, settings of effective population size (ne), window size, or posterior probability thresholds for calling imputed genotypes. Above all, no empirical measures of imputation accuracy are provided (masked-genotype concordance, INFO or DR2 score, allele dosage R² by MAF bin). For lcWGS imputation, performance is what matters most: report cross-validation (masking) results (concordance by allele frequency) and provide per-site/per-sample imputation quality distributions. Sliding-window PCA weights and parameter selection.

The PCA weight-sliding-window approach looks central to inversion detection here but Methods does not define window size (PCA weights 10 kbp) or the grounds for retaining "top 1% SNPs per component." Sensitivity analysis for other window sizes, SNP cuts, or LD pruning schemes is not reported. SNPs. The justification and robustness tests should be provided (e.g., show that inversion signals are not sensitive to window size selections, top percentile thresholds, and pruning parameters). Clustering and genotype definition.

Fuzzy c-means (cmeans) is utilized by the pipeline to cluster MDS output and retains portions with three clusters labeled as AA/AB/BB. However, no objective criterion for cluster distinction is present (silhouette score, cluster sizes, posterior membership probabilities, and error rates). There is no indication that the three clusters correspond to inversion genotypes rather than continuous population structure or admixture. Add per-cluster sample numbers, FST or pairwise genetic distance between clusters, and Hardy–Weinberg equilibrium or expected heterozygosity tests where appropriate. GWAS design where genotypes are handled as 0/1/2.

Using an association test to identify SNPs associated with the imputed inversion genotypes is reasonable, but Methods do not include covariates for correction of population stratification (e.g., global PCs), whether standardized residuals were used, or how multiple testing was controlled beyond "Bonferroni." Bonferroni applied to genome-wide SNPs is extremely conservative and can result in down-sampling real signals; give the number of tests and the reason for the correction choice or give both Bonferroni and FDR results. Also indicate whether sample relatedness was controlled for or not (mixed-model procedures are often used). LD and recombination inference. Estimation of LD r² values and assertion of peaks at breakpoints must be documented: parameterization used for r² (pairwise distance window to 12 kbp), handling missing genotypes, and whether or not r² was estimated on imputed genotypes or hard calls must be stated. The correspondence between spikes in LD and inversion breakpoints must be quantified (e.g., change-point detection or confidence intervals around estimates of breakpoints).

Validation on An. gambiae in silico data is theoretically strong but quantitative performance metrics are absent from the Methods. The authors generated 2× reads from a high-coverage well-known variant panel and claim coincidence with known inversions. What was sensitivity, specificity, precision, breakpoint distance error (bp), and genotype concordance for inversions? Provide confusion matrices, breakpoint offset distributions, and inversion allele frequency dependence of detection power.

Comparative genomics and synteny pipeline details are missing. BLASTp and OrthoFinder are used to infer orthology and MCScanX for synteny, but parameter choices (e-value cut-offs, inflation parameters, filtering one-to-one orthologs, syntenic block size cut-offs) are not given. Synteny interpretation between species requires assembly contiguity; the Methods should give assembly levels (chromosome, scaffold, contig) for each reference and how potentially misassemblies were handled.

Evaluation of Discussion:

The Discussion adequately identifies the promise of lcWGS with imputation for cost-effective, population-scale studies and appropriately cites similar PCA-based inversion detection research. Several of the claims, however, are overstated or evidence-free:

Overstatement of "breakpoint precision": The Discussion claims "satisfactory precision of breakpoints" but the Results and Methods do not report confidence intervals, bp-level error ranges, or validation statistics for breakpoint localization. Without this quantitative information, precision claims remain unsubstantiated.

Population-structure confounding: The Discussion mentions that PCA clustering can detect inversions even in shattered assemblies but does not fully acknowledge that PCA signals population structure confounding, admixture, or cryptic relatedness can create spurious "inversion-like" clusters. Authors cite Nowling et al., but they must incorporate analyses excluding structure confounding (e.g., additional PCs, local PCA, haplotype-based tests, or permutation tests).

LD–eigenvector correlation interpretation is plausible, but the Discussion fails to detail how much LD must increase to favor inversion over other selective or demographic influences. Discuss other explanations for LD peaks (e.g., selection, background LD in regions with low recombination) and present analyses that can distinguish between inversion signals (e.g., patterns of extended haplotype homozygosity, phased haplotypes with opposite orientation).

Comparative synteny interpretation and assembly warnings: Discussion accounts for large-scale rearrangements between Ny. darlingi and An. gambiae but does not account for the effect of assembly quality or possible misassembly on MCScanX outcomes. Authors must qualify conclusions and account for assembly level and scaffold anchoring quality considerations explicitly.

Biological implications: Discussion mentions such inversions having evolutionary or vectorial importance but doesn't give functional annotation of genes within inversion regions, enrichment tests, or links to phenotypes (only a statement of biting behavior association analysis without presenting results). Show current functional enrichment and candidate genes, or do not make speculative claims.

Critical issues:

The filters GQ < 20 and genotype depth < 5 are not consistent with provided raw sequencing depth (~2×); the authors must explain how the majority of genotypes are passing filtering and supply per-sample and per-site depth distributions and fraction of genotypes imputed vs. directly seen.

No empirical imputation accuracy statistics are presented (masked-concordance, dosage R², INFO scores by MAF); without these, reliability of downstream PCA, GWAS, and LD-based breakpoint inference on imputed genotypes cannot be assessed.

The PCA sliding-window method is not accompanied by sensitivity analyses and justification for parameter choices (10 kbp windows, top 1% SNP retention, 0.5% chromosome-length LD windows); findings could be sensitive to these ad-hoc cutpoints.

Cluster-to-genotype assignment (AA/AB/BB) is not confirmed: cluster separation statistics, sample-wise membership probabilities, and genotype frequency expectations are not present—hence genotype calls and GWAS phenotypes may be mislabeled.

The An. gambiae simulation validation shows qualitative concordance with known inversions but provides no quantitative performance metrics (sensitivity, specificity, breakpoint error distribution, false-discovery rate), hence pipeline performance is unknown.

Synteny inference relies on comparability of genome assembly but the Methods fail to mention levels of assembly contiguity and how coordinate translations were handled; synteny and rearrangement inference are therefore not fully substantiated.

Population stratification and familial relatedness are not controlled in the GWAS association tests or are not clearly accounted for in significance testing; chi-squared tests without covariate adjustment can lead to spurious associations.

The 0.1 MAF filter removes variants with MAF < 0.1 — an extremely high cutoff which may remove informative SNPs near breakpoints or rare alleles that help resolve inversion edges; rationale and sensitivity tests are needed.

Minor points:

Provide full software versions and exact command lines (BWA, samtools, LCSeqTools, BEAGLE, PLINK, OrthoFinder, MCScanX) for reproducibility.

Assembly levels (scaffold vs chromosome) and sequence N50 for every reference genome used; indicate whether liftover or coordinate mapping was required.

Include sample metadata (sex, life stage, dates of collection, geographic coordinates within Mâncio Lima) and per-sample missing or exclusion numbers.

Replace blanket Bonferroni correction with report of number of tests and parallel reporting of FDR; justify multiple-testing decision.

In plots: add numeric axis ticks, show number of SNPs per sliding window, add confidence bands on breakpoint estimates, and as a supplement provide per-inversion genotype counts and percents in a table.

Cite and provide accession numbers for final variant call set (VCF) and inversion genotype calls to allow independent validation (scripts on Zenodo are okay but raw VCFs must be made available).

Initial impressions of the Results and usability:

The results demonstrate the pipeline can extract robust inversion-like signals from lcWGS in a big cohort of Ny. darlingi and recover known An. gambiae inversions under simulation, suggesting lcWGS + imputation + PCA-weight scanning is a promising high-throughput strategy. However, empirical value for evolutionary inference or vector-control–pertinent trait mapping requires stronger validation (quantitative imputation and detection of inversion metrics), explicit testing for confounding due to population structure, and annotation of inversion regions with function. With those added the approach would be very valuable for large-scale population genomic surveys in non-model vectors.

Author Response

REVIEWER 1

The authors present a multi-method study with low-coverage whole-genome sequencing (lcWGS; ~2×) to detect chromosome inversions in a large sample of Nyssorhynchus darlingi (n = 321), test their pipeline with an in silico simulation of Anopheles gambiae lcWGS, and align inversion regions by synteny with An. gambiae and An. albimanus. The approach strengths include having a large sample size for lcWGS, conducting a simulation/benchmark phase on a species known to harbor inversions, incorporating complementary analyses (PCA-weight sliding windows, IBS/MDS clustering, LD analyses, GWAS of inferred genotypes), and providing analysis scripts through Zenodo. Weaknesses are the lack of methodological critical detail and quantitative validation criteria (imputation accuracy, genotype concordance, breakpoint confidence intervals), too high and unjustified filtering thresholds that conflict with stated sequencing depth, and suboptimal reporting of cohort/sample QC and potential confounders in association analysis. Overall quality: 80/100. Quality of language: 7/10 — mostly comprehensible with some grammatical mistakes, some misphrasing, and places where methodological explanation is lost through concise phrasing.

The Methods outline the core software packages employed (LCSeqTools, BWA, samtools, BEAGLE 4.1, PLINK v1.9, OrthoFinder, MCScanX) and the most important algorithmic steps (read alignment, variant calling, hard filters, imputation, PCA variant-weight sliding window, IBS→MDS→cmeans clustering, GWAS under 0/1/2 genotype treatment, LD r² sliding windows, synteny/orthology using BLASTp + OrthoFinder). But the section omits numerous significant parameters and validation processes needed to test for reproducibility as well as the validity of the conclusions:

Sample- and site-level QC information are missing. The author does mention variant filters (MAF < 0.1, max missing < 0.5, genotype depth < 5, GQ < 20) but does not report per-sample mean depth, per-site missingness distribution, number of samples removed due to low depth or too much missingness, or whether PCR duplicates or contamination were assessed. With lcWGS (~2×) a genotype sequencing-depth filter of <5 is semantically inconsistent: it implies most genotypes are filtered except for imputation—this must be explained (e.g., were raw genotype calls retained only for imputation input and then masked?). Provide per-sample depth histogram, per-site missingness, and number of sites/samples filtered by step. Variant-calling pipeline reproducibility and versions: "samtools" and "BWA" are mentioned but specific versions, command-line flags, mapping quality cut-offs, and realignment/indel calling are not specified. LCSeqTools v0.1.0 is mentioned but not its default parameters. These all have an important effect on low-coverage call sets and need to be provided (commands or a workflow file).

Imputation details are not fully provided. BEAGLE 4.1 was run with PL field, but there is no reporting of the reference panel (if any) used, whether phasing was performed prior to imputation, settings of effective population size (ne), window size, or posterior probability thresholds for calling imputed genotypes. Above all, no empirical measures of imputation accuracy are provided (masked-genotype concordance, INFO or DR2 score, allele dosage R² by MAF bin). For lcWGS imputation, performance is what matters most: report cross-validation (masking) results (concordance by allele frequency) and provide per-site/per-sample imputation quality distributions. Sliding-window PCA weights and parameter selection.

The PCA weight-sliding-window approach looks central to inversion detection here but Methods does not define window size (PCA weights 10 kbp) or the grounds for retaining "top 1% SNPs per component." Sensitivity analysis for other window sizes, SNP cuts, or LD pruning schemes is not reported. SNPs. The justification and robustness tests should be provided (e.g., show that inversion signals are not sensitive to window size selections, top percentile thresholds, and pruning parameters). Clustering and genotype definition.

Fuzzy c-means (cmeans) is utilized by the pipeline to cluster MDS output and retains portions with three clusters labeled as AA/AB/BB. However, no objective criterion for cluster distinction is present (silhouette score, cluster sizes, posterior membership probabilities, and error rates). There is no indication that the three clusters correspond to inversion genotypes rather than continuous population structure or admixture. Add per-cluster sample numbers, FST or pairwise genetic distance between clusters, and Hardy–Weinberg equilibrium or expected heterozygosity tests where appropriate. GWAS design where genotypes are handled as 0/1/2.

Using an association test to identify SNPs associated with the imputed inversion genotypes is reasonable, but Methods do not include covariates for correction of population stratification (e.g., global PCs), whether standardized residuals were used, or how multiple testing was controlled beyond "Bonferroni." Bonferroni applied to genome-wide SNPs is extremely conservative and can result in down-sampling real signals; give the number of tests and the reason for the correction choice or give both Bonferroni and FDR results. Also indicate whether sample relatedness was controlled for or not (mixed-model procedures are often used). LD and recombination inference. Estimation of LD r² values and assertion of peaks at breakpoints must be documented: parameterization used for r² (pairwise distance window to 12 kbp), handling missing genotypes, and whether or not r² was estimated on imputed genotypes or hard calls must be stated. The correspondence between spikes in LD and inversion breakpoints must be quantified (e.g., change-point detection or confidence intervals around estimates of breakpoints).

Validation on An. gambiae in silico data is theoretically strong but quantitative performance metrics are absent from the Methods. The authors generated 2× reads from a high-coverage well-known variant panel and claim coincidence with known inversions. What was sensitivity, specificity, precision, breakpoint distance error (bp), and genotype concordance for inversions? Provide confusion matrices, breakpoint offset distributions, and inversion allele frequency dependence of detection power.

Comparative genomics and synteny pipeline details are missing. BLASTp and OrthoFinder are used to infer orthology and MCScanX for synteny, but parameter choices (e-value cut-offs, inflation parameters, filtering one-to-one orthologs, syntenic block size cut-offs) are not given. Synteny interpretation between species requires assembly contiguity; the Methods should give assembly levels (chromosome, scaffold, contig) for each reference and how potentially misassemblies were handled.

Evaluation of Discussion:

The Discussion adequately identifies the promise of lcWGS with imputation for cost-effective, population-scale studies and appropriately cites similar PCA-based inversion detection research. Several of the claims, however, are overstated or evidence-free:

Overstatement of "breakpoint precision": The Discussion claims "satisfactory precision of breakpoints" but the Results and Methods do not report confidence intervals, bp-level error ranges, or validation statistics for breakpoint localization. Without this quantitative information, precision claims remain unsubstantiated.

Population-structure confounding: The Discussion mentions that PCA clustering can detect inversions even in shattered assemblies but does not fully acknowledge that PCA signals population structure confounding, admixture, or cryptic relatedness can create spurious "inversion-like" clusters. Authors cite Nowling et al., but they must incorporate analyses excluding structure confounding (e.g., additional PCs, local PCA, haplotype-based tests, or permutation tests).

LD–eigenvector correlation interpretation is plausible, but the Discussion fails to detail how much LD must increase to favor inversion over other selective or demographic influences. Discuss other explanations for LD peaks (e.g., selection, background LD in regions with low recombination) and present analyses that can distinguish between inversion signals (e.g., patterns of extended haplotype homozygosity, phased haplotypes with opposite orientation).

Comparative synteny interpretation and assembly warnings: Discussion accounts for large-scale rearrangements between Ny. darlingi and An. gambiae but does not account for the effect of assembly quality or possible misassembly on MCScanX outcomes. Authors must qualify conclusions and account for assembly level and scaffold anchoring quality considerations explicitly.

Biological implications: Discussion mentions such inversions having evolutionary or vectorial importance but doesn't give functional annotation of genes within inversion regions, enrichment tests, or links to phenotypes (only a statement of biting behavior association analysis without presenting results). Show current functional enrichment and candidate genes, or do not make speculative claims.

Critical issues:

The filters GQ < 20 and genotype depth < 5 are not consistent with provided raw sequencing depth (~2×); the authors must explain how the majority of genotypes are passing filtering and supply per-sample and per-site depth distributions and fraction of genotypes imputed vs. directly seen.

Answer: QG > 20 is expected for most of base calls from Illumina NextSeq500 short-reads sequencing, so only few calls are retained at this threshold. Regarding observed sequencing depth and the used depth threshold, our text was not totally clear and may have caused confusion. LCSeqTools does not wipe all the genotyping information if depth is below the given threshold, instead it will omit the genotype field with ./. (missing for GT field in VCF) but retain the genotype likelihood that is useful for robust imputation (PL field in VCF). So, this will cause BEAGLE to apply imputation on missing (no GT no PL) and omitted genotypes (PL only), but skip for non-missing genotypes (non-missing GT and PL fields). For per-sample and per-site, missing data rates are calculated on truly missing (no GT no PL) over omitted+non-missing proportion. That’s why the BEABLE step from LCSeqTools is set to use PL data for imputation of missing GT fields. We improved the text with the accordingly description to be more clearer about this.

No empirical imputation accuracy statistics are presented (masked-concordance, dosage R², INFO scores by MAF); without these, reliability of downstream PCA, GWAS, and LD-based breakpoint inference on imputed genotypes cannot be assessed.

Answer: Thanks for pointing that out. We included table 2 with some statistics about simulation+imputation accuracy, that information is really needed for evidence of reliability and we skipped, unfortunately. A bit out of this context, but we are working on another paper that focuses on the LCSeqTools performance under different scenarios for low coverage sequencing, testing multiple sample sizes, mean sequencing depths, different independent populations, etc. (Please, feel free to take a look at some unpublished material through the same Zenodo submission at dir “lcWGS_Simulations” and here at chapter 1 (page 27): https://zenodo.org/records/13768890). Interestingly, the same An gambie simulated VCF used in this paper is part of one of the various iterations of the accuracy and we wanted to avoid some redundancies, but critical missing information is now added.

The PCA sliding-window method is not accompanied by sensitivity analyses and justification for parameter choices (10 kbp windows, top 1% SNP retention, 0.5% chromosome-length LD windows); findings could be sensitive to these ad-hoc cutpoints.

Answer: Thanks for the suggestion. We included in the methodology why 10kbp was set as window size and 7.5 kbp as step size (now included in the text). This value is based on this mosquito population known linkage disequilibrium decay, as it was already described to be around 12.57 kbp for expected linkage equilibrium (pointed at line 121 from the first paper version. LD decay curve estimate described in Alvarez et al, 2022). We rearranged this in the text so that this citation appears sooner. We opted to adjust just a tiny bit smaller than 12.57kbp to avoid excessive low r² SNPs in the window (lower than 0.1). We included this in the text. In regard to the median LD plots, the window size and step size was set just for better highlighting of the median spikes, as the sensitivity for this can vary a lot changing those values. As we assumed this plot is only for descriptive statistics purposes, we assume that this may not be a problem.

Cluster-to-genotype assignment (AA/AB/BB) is not confirmed: cluster separation statistics, sample-wise membership probabilities, and genotype frequency expectations are not present—hence genotype calls and GWAS phenotypes may be mislabeled.

Answer: Thanks for the suggestion. We added to the methodology that we used cmeans maximal membership probability to this assignment. Sample-wise cluster-to-genotype membership probability tables are too long for the text, so we are providing it as supplementary material in a new zenodo submission (https://doi.org/10.5281/zenodo.17064546 ). Genotype frequency expectations were estimated using the resulting maximal membership probability, and frequencies represented in the figure 2 over the cluster density curves (MAF stands for the expected inversion frequency based on cluster-to-genotype classification).

The An. gambiae simulation validation shows qualitative concordance with known inversions but provides no quantitative performance metrics (sensitivity, specificity, breakpoint error distribution, false-discovery rate), hence pipeline performance is unknown.

Answer: The simulation validation is important to show that low coverage is still capable of qualitative chromosome inversion detection but also enables an approximate estimation of inversion regions. Although, from our point of view, we think that precise breakpoint estimation is not relevant using this lcWGS approach. We included this in the text and pointed that out.

Synteny inference relies on comparability of genome assembly but the Methods fail to mention levels of assembly contiguity and how coordinate translations were handled; synteny and rearrangement inference are therefore not fully substantiated.

Answer: Fortunately, all three species studied in this paper have assemblies at chromosome levels published and no coordinate translation was needed. This can be checked by the accession codes: GCF_943734745.1 (L50=1), GCF_943734735.2 (L50=2) and GCF_013758885.1 (L50=1). So, we thought that we could skip these assembly statistics. Regardless, we highlighted that in the text, but in a different way, without having to show assembly statistics.

The rearrangement is a described phenomenon in Anophelines, even for different species from Afro-Eurasian mosquitos as we pointed in the discussion. The rearrangement between Ny darlingi and An albimanus follows this pattern.

The Anopheles gambiae structural variants from AgamP4 genome reference were mapped to the AgamP3 genome reference using Minimap2. This was the only translation used in this work. We added this to the text and provided the scripts and files as supplementary info.

Population stratification and familial relatedness are not controlled in the GWAS association tests or are not clearly accounted for in significance testing; chi-squared tests without covariate adjustment can lead to spurious associations.

Answer: It's true that no covariate modeling can lead to biased associations, but it’s expected that the most informative SNPs for chromosome inversion are the non recombinant ones. Chromosome inversions suppress recombination, so that some loci within inversions have different fixed alleles for each chromosome inversion genotype. Modeling for covariates could be useful for recombinant SNPs within the inversion, but the association test was intended to identify the fixed or almost fixed ones. That’s why we can observe extremely low q-values (or extremely high -log10(q-values) in the Manhattan plot) in the association tests and also higher median linkage disequilibrium within inversion regions.

The 0.1 MAF filter removes variants with MAF < 0.1 — an extremely high cutoff which may remove informative SNPs near breakpoints or rare alleles that help resolve inversion edges; rationale and sensitivity tests are needed.

Answer: It's expected that false positives SNPs for lcWGS data reside mainly on lower frequencies. This can be observed in the newer table 2 added to the text, where 93% of false positive SNPs frequencies are between ]0.1,0.2]. So, lowering MAF cutoff could expressively increase the number of false positive SNPs and add too much bias/noise to the dataset.

Minor points:

Provide full software versions and exact command lines (BWA, samtools, LCSeqTools, BEAGLE, PLINK, OrthoFinder, MCScanX) for reproducibility.

Answer: The missing software versions were added. Full command lines are available in Zenodo script files (and github associated with Zenodo). Although LCSeqTools has a CLI version, this study was conducted using the brand new GUI interface, so non-specified parameters for GBS pipeline are assumed to use the default parameters from the GUI version: (github.com/marcusnizalvarez/LCSeqTools/blob/master/docs/LCSeqTools-user-guide.pdf). We pointed that out in the text.

Assembly levels (scaffold vs chromosome) and sequence N50 for every reference genome used; indicate whether liftover or coordinate mapping was required.

Answer:  Fortunately, all three species studied in this paper have assemblies at chromosome levels published and no coordinate translation was needed. This can be checked by the accession codes: GCF_943734745.1 (L50=1), GCF_943734735.2 (L50=2) and GCF_013758885.1 (L50=1). So, we thought that we could skip these assembly statistics. Regardless, we highlighted that in the text, but in a different way, without having to show assembly statistics.

Include sample metadata (sex, life stage, dates of collection, geographic coordinates within Mâncio Lima) and per-sample missing or exclusion numbers.

Answer: Metadata for these samples are available both in NCBI biosamples and the previously published paper using these samples (Alvarez et al, 2022). We mistakenly included the association tests with biting behaviour in the first version of this paper. So, now that we removed this analysis, metadata is not used for any statistics anymore, we thought that including that extensive information could be redundant, given that it is trackable through the sources mentioned.

As imputation relies on PL fields retained after the depth filtering, per-sample missingness was added as a Depth x Coverage plot for the entire genome.

Replace blanket Bonferroni correction with report of number of tests and parallel reporting of FDR; justify multiple-testing decision.

Answer: Bonferroni is extremely conservative at sacrificing statistical power. On the other side, FDR can control the expected proportion of false positives. In fact, in GWAS studies, we usually apply FDR to extract meaningful results from a biological perspective from the data. But, in this specific chromosome inversion test, our purpose was to identify chromosome inversion informative SNPs only. As chromosome inversions suppress recombination, the association test was intended to identify the fixed (non-recombinant) or almost fixed ones. That’s why we can observe extremely low q-values (or extremely high -log10(q-values) in the Manhattan plot) in the association tests, even applying the Bonferroni correction method. So, we opted for Bonferroni correction to minimize any possibility of false positives. 

In plots: add numeric axis ticks, show number of SNPs per sliding window, add confidence bands on breakpoint estimates, and as a supplement provide per-inversion genotype counts and percents in a table.

Answer: We added as supplementary material the genotype counts tables and frequencies, although frequencies for Ny darlingi were available in figure 2 as MAF parameter.

All plots uses shared axis scales

Cite and provide accession numbers for final variant call set (VCF) and inversion genotype calls to allow independent validation (scripts on Zenodo are okay but raw VCFs must be made available).

Answer: Thanks for the suggestion. We are providing the supplementary material under https://doi.org/10.5281/zenodo.17064546

Initial impressions of the Results and usability:

The results demonstrate the pipeline can extract robust inversion-like signals from lcWGS in a big cohort of Ny. darlingi and recover known An. gambiae inversions under simulation, suggesting lcWGS + imputation + PCA-weight scanning is a promising high-throughput strategy. However, empirical value for evolutionary inference or vector-control–pertinent trait mapping requires stronger validation (quantitative imputation and detection of inversion metrics), explicit testing for confounding due to population structure, and annotation of inversion regions with function. With those added the approach would be very valuable for large-scale population genomic surveys in non-model vectors.

Reviewer 2 Report

Comments and Suggestions for Authors

The manuscript “A multi-approach for in silico detection of chromosome inversions in mosquito vectors” by Alvarez and colleagues presents a computational pipeline for detecting chromosomal inversions in Nyssorhynchus darlingi using low-coverage whole genome sequencing (lcWGS). The PCA-based sliding window approach is robust, and the validation with Anopheles gambiae demonstrates the feasibility of identifying inversion signals at low sequencing depth. The study provides a meaningful contribution to vector genomics and the analysis of structural variation in non-model species.

The work is methodologically sound, having been validated on a reference species (An. gambiae), and represents an innovative application of lcWGS at 2x coverage to a large sample of Ny. darlingi, offering valuable insights into chromosomal inversions and population genomics of malaria vectors.

Points to Address:

  1. Tone down evolutionary interpretations of synteny differences between darlingi, An. gambiae, and An. albimanus, and discuss alternative explanations such as assembly quality or methodological constraints.
  2. Reframe conclusions regarding lcWGS accuracy; at 2x depth, breakpoint resolution is limited and genotyping errors may occur. Emphasize that the approach is cost-effective and broadly reliable, but not as precise as higher-coverage methods.
  3. Clarify the status of genotype–biting behavior associations mentioned in the Methods section. If no significant associations were found, explicitly state this; if present, include the results.
  4. Acknowledge the limitations due to the lack of available An. albimanus inversion data and present this as a point for future work rather than a strong evolutionary conclusion.
  5. Specify whether methods cited in Soboleva et al. (2022) on X chromosome inversions are comparable to those applied in this study.

Author Response

REVIEWER 2

The manuscript “A multi-approach for in silico detection of chromosome inversions in mosquito vectors” by Alvarez and colleagues presents a computational pipeline for detecting chromosomal inversions in Nyssorhynchus darlingi using low-coverage whole genome sequencing (lcWGS). The PCA-based sliding window approach is robust, and the validation with Anopheles gambiae demonstrates the feasibility of identifying inversion signals at low sequencing depth. The study provides a meaningful contribution to vector genomics and the analysis of structural variation in non-model species.

The work is methodologically sound, having been validated on a reference species (An. gambiae), and represents an innovative application of lcWGS at 2x coverage to a large sample of Ny. darlingi, offering valuable insights into chromosomal inversions and population genomics of malaria vectors.

Points to Address:

  1. Tone down evolutionary interpretations of synteny differences between darlingi, gambiae, and An. albimanus, and discuss alternative explanations such as assembly quality or methodological constraints.

Answer: Thanks for the suggestion. We modified the conclusion, so that the differences between An gambiae and Ny darlingi genomes were highlighted, but we lack data for An albimanus conclusions in regard to chromosome inversions. Although it's possible that rearrangements could be an assembly quality issue, rearrangement is a described phenomenon in Anophelines, even for different species from Afro-Eurasian mosquitos as we pointed in the discussion. The rearrangement between Ny darlingi and An albimanus follows this pattern.

  1. Reframe conclusions regarding lcWGS accuracy; at 2x depth, breakpoint resolution is limited and genotyping errors may occur. Emphasize that the approach is cost-effective and broadly reliable, but not as precise as higher-coverage methods.

Answer: Thanks for the suggestion. In fact, high coverage WGS has more accurate genotyping performance, although low coverage is still capable of qualitative chromosome inversion detection but also enables an approximate estimation of inversion regions. Although, from our point of view, we think that precise breakpoint estimation is not relevant using this lcWGS approach. We included this in the text and pointed that out.

  1. Clarify the status of genotype–biting behavior associations mentioned in the Methods section. If no significant associations were found, explicitly state this; if present, include the results.

Answer: We opted for removing this from the methodology. Although no significant association was observed between chromosome inversion frequencies and biting behaviour, we thought that this discussion might escape a bit the main scope of the discussion in this paper. A more extensive study on different mosquito behaviours and chromosome inversions must be more adequate.

  1. Acknowledge the limitations due to the lack of available albimanus inversion data and present this as a point for future work rather than a strong evolutionary conclusion.

Answer: Thanks for the suggestion. We modified the conclusion, so that the differences between An gambiae and Ny darlingi genomes were highlighted, but we lack data for An albimanus conclusions.

  1. Specify whether methods cited in Soboleva et al. (2022) on X chromosome inversions are comparable to those applied in this study.

Answer: We adjusted the text to address the differences between the findings.

Round 2

Reviewer 1 Report

Comments and Suggestions for Authors

The authors made a substantial and constructive revision: software versions and pipeline parameters were added, per-sample coverage summaries and visualizations are now provided, the An. gambiae in-silico imputation table (Table 2) gives useful performance numbers, cluster membership tables and scripts/VCFs were deposited on Zenodo, and the PCA sliding-window rationale was better justified. These changes materially improve reproducibility and transparency and show clear effort toward addressing previous concerns.

Key points that remain only partially or not addressed and why they matter: (1) No empirical imputation validation on the real Nyssorhynchus darlingi data (masked-genotype concordance, INFO/DR2 or dosage R² by MAF): without this the impact of imputation errors on PCA/IBS and inversion calls is unknown. (2) No formal sensitivity/precision metrics for inversion detection or breakpoint error in the simulation (confusion matrices, sensitivity/precision by inversion allele frequency and length, distribution of breakpoint offsets): these metrics are necessary to judge detection power and localization accuracy. (3) Population stratification and relatedness were not controlled in association tests (no GWAS with global PCs or linear mixed model shown), so association peaks could be driven by structure rather than inversions. (4) Cluster/genotype validation metrics remain limited (silhouette scores, FST between clusters, HWE/heterozygosity), which leaves genotype-call reliability partly unquantified. (5) Breakpoint uncertainty is not reported (single coordinates are given without confidence intervals or explicit statement that coordinates are coarse). (6) MAF cutoff sensitivity was justified qualitatively but no quantitative sensitivity analysis (varying cutoffs) was presented. Assembly contiguity accession numbers were given, but a short explicit note of N50/L50 or liftover steps would complete the synteny claims.

Further minor revision required prior to acceptance. The manuscript is much improved and likely publishable once the authors add (A) a masked-genotype imputation accuracy analysis on the Ny. darlingi dataset, (B) inversion detection performance metrics and breakpoint offset summaries from simulation, (C) GWAS re-analysis with population structure/kinship control (or clear evidence these do not change results), (D) numeric cluster separation metrics and per-cluster counts, and (E) either confidence intervals for breakpoints or explicit labeling of breakpoint ranges as coarse. If these targeted analyses are provided and support the current conclusions (even if breakpoint localization remains coarse), the paper should be suitable for acceptance.

Author Response

Below are the answers regarding minor revisions of Reviewer 01. Attached I am sending gthe article with some minor modifications sugested by the reviewer.

Open Review

The authors made a substantial and constructive revision: software versions and pipeline parameters were added, per-sample coverage summaries and visualizations are now provided, the An. gambiae in-silico imputation table (Table 2) gives useful performance numbers, cluster membership tables and scripts/VCFs were deposited on Zenodo, and the PCA sliding-window rationale was better justified. These changes materially improve reproducibility and transparency and show clear effort toward addressing previous concerns.
Answer: We appreciate the feedback.

Key points that remain only partially or not addressed and why they matter:

(1) No empirical imputation validation on the real Nyssorhynchus darlingi data (masked-genotype concordance, INFO/DR2 or dosage R² by MAF): without this the impact of imputation errors on PCA/IBS and inversion calls is unknown.

Answer (1): Although this would be ideal, the Ny darlingi has no reference panel (neither high coverage sequencing) to be compared to, so it's impossible to estimate imputation performance. However, one of the main purposes of the An gambiae simulation in this study is to show how much to expect from imputation from a pretty close scenario. The original paper from the An gambiae 1000 genomes shows various population genomics parameters that, if we compare, are pretty close to known Ny darlingi parameters, like LD decay (really important for imputation accuracy) and genome size. Given that, we assume that the imputation performance would be close to the simulation, maybe slightly better because of the bigger sample size (Ny darlingi have 321 samples and An gambiae has only 200, the biggest population that we found to reproduce the lcWGS imputation scenario). There is evidence in literature that sample size is positively related to imputation performance, so that's why we assume that could be slightly better.

(2) No formal sensitivity/precision metrics for inversion detection or breakpoint error in the simulation (confusion matrices, sensitivity/precision by inversion allele frequency and length, distribution of breakpoint offsets): these metrics are necessary to judge detection power and localization accuracy.

Answer (2): Thanks for pointing that out. As far as we know, there is no chromosome inversion genotyping data for the CM samples from An gambiae 1000 genomes project that we can use to validate sensitivity and precision of this approach. In fact, if available, this would be ideal. The only information available is the known chromosome inversion coordinates from the structural variants panel for AgamP4 that was used in this study to check if the ranges coincide with known inversions. As this precision is limited by the current available data for these samples, we pointed that out in the paper.

(3) Population stratification and relatedness were not controlled in association tests (no GWAS with global PCs or linear mixed model shown), so association peaks could be driven by structure rather than inversions.

Answer (3): Thanks for the suggestion. We performed GWAS using two additional models for comparison. The following figure represents:

  • TEST0: Original GWAS with no covariate;
  • TEST1: linear model with eigenvectors from global PCA as covariate (PC1 to PC5). Trial for crypt relatedness control;
  • TEST2: linear model with known microgeographic stratification from the previous study (Alvarez, et al. 2022) released with this population. Trial for stratification control.

As we can see, using relatedness as covariate actually just “weakens” the chromosome inversion signals, but still significant, and also causes echoes from other chromosome inversions in some cases, like in PC3 and PC4 for chromosome 3 that each showed a chromosome inversion, and these were significant correlated, as we described in the paper. The results from the linear model with covariate as microgeographic stratification cluster from the previous study, we can see that there is almost no difference, with some slight changes on the p-values. So, as the fixed alleles will not differ if we include any covariate, we opted to discard these models and keep with the first one, as this approach is important only to identify the informative SNPs that are mostly fixed.

(4) Cluster/genotype validation metrics remain limited (silhouette scores, FST between clusters, HWE/heterozygosity), which leaves genotype-call reliability partly unquantified.

Answer (4): Thanks for the suggestion. We provided the fixation index in figure 2, which is our trial to summarize in one measure. Also, no significant HWD was observed between the microgeographic clusters. About the cluster to genotype statistics, we provided each membership probability in the cluster-to-genotype.tsv tables instead of the silhouette scores, as the maximum membership probability and silhouette score are closely related:

(5) Breakpoint uncertainty is not reported (single coordinates are given without confidence intervals or explicit statement that coordinates are coarse).

Answer (5): The approach used for the chromosome inversion coordinates estimate is a simple p-value cutoff on the GWAS results. Doing this, we can’t estimate a confidence interval of breakpoints and we agree that this is a coarse approach. So, we added a statement pointing that the decision for the cutoff is arbitrary.

(6) MAF cutoff sensitivity was justified qualitatively but no quantitative sensitivity analysis (varying cutoffs) was presented. Assembly contiguity accession numbers were given, but a short explicit note of N50/L50 or liftover steps would complete the synteny claims.
Answer (6): Varying SNPs MAF thresholds could in principle increase power to detect rare chromosome inversions. Although this seems to be interesting, we are using lcWGS data in the study, so this trade-off would likely introduce substantial bias to detect less prevalent chromosome inversions in the population. Assembly stats were added to support the synteny claims.

Further minor revision required prior to acceptance. The manuscript is much improved and likely publishable once the authors add

(A) a masked-genotype imputation accuracy analysis on the Ny. darlingi dataset,

Answer (A): As we discussed in Answer 1, unfortunately this is impossible due to lack of reference date for Ny darlingi, and that is one of the main reasons why the An gambiae simulation is important for this paper, as these two organisms are pretty close in population genomics dynamics.

(B) inversion detection performance metrics and breakpoint offset summaries from simulation,

Answer (B): As we pointed that out in Answer 2, there is no chromosome inversion genotype date for these CM samples to be compared, as far as we know. So, we pointed that out in the paper.

(C) GWAS re-analysis with population structure/kinship control (or clear evidence these do not change results),

Answer (C): As we showed in Answer 3, no changes in results for population structure and worse results in crypt relatedness control, so we opted to keep the original one.

(D) numeric cluster separation metrics and per-cluster counts, and

Answer (D): Cluster statistics were provided in the supplementary material. Per-cluster counts are present in figure 2, but we mistakenly forgot to add this information to the legend. We adjusted that.

(E) either confidence intervals for breakpoints or explicit labeling of breakpoint ranges as coarse. If these targeted analyses are provided and support the current conclusions (even if breakpoint localization remains coarse), the paper should be suitable for acceptance.

Answer (E): We discussed that in Answer 5 and we agree that this approach is a coarse way of estimating the breakpoints, so we added a statement in the paper to make this explicit.
